# Effectiveness of Myofunctional Therapy in Ankyloglossia: A Systematic Review

**DOI:** 10.3390/ijerph191912347

**Published:** 2022-09-28

**Authors:** María del Puerto González Garrido, Cristina Garcia-Munoz, Manuel Rodríguez-Huguet, Francisco Javier Martin-Vega, Gloria Gonzalez-Medina, Maria Jesus Vinolo-Gil

**Affiliations:** 1Department of Nursing and Physiotherapy, University of Cadiz, 11003 Cadiz, Spain; 2Rehabilitation Clinical Management Unit, Interlevels-Intercenters Hospital Puerta del Mar, Hospital Puerto Real, Cadiz Bay-La Janda Health District, 11006 Cadiz, Spain; 3Biomedical Research and Innovation Institute of Cadiz (INiBICA), Research Unit, Puerta del Mar University Hospital, University of Cadiz, 11003 Cadiz, Spain; 4CTS-986 Physical Therapy and Health (FISA), University Institute of Research in Social Sustainable Development (INDESS), 11003 Cadiz, Spain

**Keywords:** ankyloglossia, tongue-tie, short lingual frenulum, myofunctional therapy

## Abstract

Ankyloglossia is a pathology of the tongue in which the frenulum appears anchored to the floor of the mouth. The treatment of choice for this pathology is frenectomy, but myofunctional therapy is emerging in recent years as a complement to surgical intervention. This systematic review aims to synthesize the scientific evidence and assess its quality regarding the use of myofunctional therapy in ankyloglossia. The Cochrane Central Register of Controlled Trials, Physiotherapy Evidence Database, Pubmed, Web of Science and Scopus were searched. Study quality was determined using the PEDro scale, STROBE statement and single-case experimental design scale. Eleven studies were selected. Based on the studies included in this review, surgery is more effective than myofunctional therapy, although better results are achieved if both are combined. Improvements have been found in maternal pain, weight gain of babies, duration of breastfeeding, tongue mobility, strength and endurance, sleep apnea, mouth breathing and snoring, quality of life, clenching teeth, myofascial tension, pain after surgery and speech sound production. These findings must be taken with caution because of the small number of articles and their quality. Future clinical trials using larger sample sizes and with higher methodological quality are needed.

## 1. Introduction

The term ankyloglossia comes from the Greek word meaning “tongue-tie”. It is a congenital anomaly in which the lingual frenulum restricts the mobility of the tongue [1]. It is most common in newborns, being more frequent in males because of its X-linked genetic characteristics [2] caused by mutations in the TBX22 gene [3].

Its prevalence ranges from 0.1 to 12%. Regarding the current frequency of diagnosis of ankyloglossia compared to 20 years before, there has been an increase probably related to concerns about its impact on breastfeeding [4]. The World Health Organisation (WHO) recommends that babies should be breastfed for the first six months and continue until two years of age, along with food intake [5].

Its main consequence is difficulty in breastfeeding because of ineffective tongue movements that lead to poor nipple attachment and sucking, causing pain and cracks and hindering milk extraction, affecting the mother and the infant’s development [4].

Other disorders in adults have also been linked to ankyloglossia, such as solid fee-ding problems, choking, nausea, feeding frustration, tongue thrusting, speech difficulties and airway obstruction [6].

Patients presenting with this pathology are often offered a frenectomy, a surgical intervention that removes the frenulum [7], which sometimes presents postoperative complications such as infections, tongue biting and bleeding [8]. There are different surgical procedures such as frenotomy, frenulectomy, frenuloplasty, miofrenuloplasty, Z-plasty and V-Y plasty [9]. Most commonly, a simple frenotomy with scissors or a laser frenectomy is performed [10].

Scientific literature supports early frenotomy in severe cases of ankyloglossia, but debate continues about treatment of mild and moderate degrees of tongue-tie resulting in extensive variations in clinical practice [11].

Physiotherapy has been used both preoperatively and postoperatively to improve the prognosis. Techniques used have included speech exercises, oral cavity morphology awareness and myofunctional therapy (MFT) involving stretching, exercises, extrabuccal and intrabuccal massages [12]. MFT can lead to the release of tongue-tie through intraoral and extraoral stimulation without the need for surgery [8].

This paper aims to provide an updated perspective on MFT research in ankyloglossia and to analyze its efficacy, as an adjunct or not to frenectomy, in improving patients with ankyloglossia. 

## 2. Materials and Methods

The PRISMA (preferred reporting items for systematic reviews and meta-analyses) [13] guidelines were followed to perform this systematic review (Appendix A). The search protocol was registered in the PROSPERO database of prospectively registered systematic reviews (CRD42022333529). 

### 2.1. Search Strategy

The literature search was performed between May and June 2022 in the following electronic databases: PubMed, Cochrane Central Register of Controlled Trials (CEN-TRAL), Scopus, PEDro (Physiotherapy Evidence Database) and Web of Science (WOS). The following descriptor terms combined with Boolean operators were employed: (“ankyloglossia” OR “tongue tie” OR “lingual frenum” OR “lingual frenulum” OR “short lingual frenulum”) AND (“myofunctional therapy” OR “tongue orofacial exercises” OR “functional therapy protocol” OR “functional therapy” OR “oral myofunctional” OR “orofacial myofunctional“ OR “myofunctional” OR “myofunctional training” OR “orofacial myo-logy” OR “orofacial myofunctional therapy” OR “therapy” OR “physiotherapy”. No date and language filters were applied. Table 1 shows the different search combinations.

### 2.2. Selection Criteria

The PICOS (population, intervention, comparison, outcomes and study design) model [14] was used to establish the inclusion criteria: (1) population: individuals of any age with ankyloglossia; (2) intervention: MFT, used as an adjunct to surgery or as a treatment; (3) comparison: absence of treatment, surgery or other therapies used; (4) outcome: any physical variable susceptible to improvement after MFT; (5) study design: controlled clinical trials, observational studies and case studies. Articles in which participants were people with disorders of the tongue, the origin of which was not ankyloglossia, were excluded. 

### 2.3. Study Selection Process and Data Extraction

The papers were independently reviewed and selected by two of the researchers (M.J.V.-G and F.J.M.-V.). The final result was agreed with a third investigator (G.G.-M.). This review and the selection were conducted in July 2022.

The information was extracted from each study related to authors, date of publication, type of study, number and gender of the sample, interventions, outcome measures, measurement instrument and results obtained.

### 2.4. Assessment of Methodological Quality

The PEDro scale was used to assess the methodological quality of the randomized clinical trials included in the review. It consists of 11 items related to the domains of selection, performance, detection, reporting and attribution bases [15]. A study with a score of 6 or more is considered as a level of evidence 1 (6–8 would be good, 9–10 would be excellent), and a study with a score of 5 or less is considered level of evidence 2 (4–5 would be acceptable, <4 would be poor) [16].

To assess the methodological quality of the observational studies, the STROBE statement was used: it contains a total of 22 items that evaluate elements such as article title, abstract, introduction, methods, results, discussion sections and other information [17].

For the evaluation of the case studies was used the single-case experimental design scale (SCED) with 11 items. Scoring ranges from 0 to 10, with higher scores suggesting higher methodological quality [18].

### 2.5. Risk of Bias of Randomised Clinical Trials

The Cochrane collaboration tool [19] and the Review Manager 5.3 software (The Cochrane Collaboration, London, UK), which includes a description and rating of each item in relation to the bias table, were used to assess the risk of bias of randomized clinical trials. After assessing the risk of bias of each study, they were classified as low risk, high risk and unclear risk. Two reviewers (M.J.V.-G. and F.J.M.-V.) assessed them independently. In case of doubt, the final decision was made through discussions with a third expert (G.G.-M.). The following types of bias were assessed: selection bias, performance bias, detection bias, attrition bias, reporting bias and other bias. 

## 3. Results

### 3.1. Selection of Studies

A total of 652 potentially relevant articles were retrieved after the selection process. The entire selection process in the different phases is detailed in a PRISMA flow chart (Figure 1).

### 3.2. Data Extraction 

A total of 11 studies were included in the systematic review, 3 case reports [20,21,22], 5 observational studies [6,8,23,24,25] and 3 randomized clinical trials [12,26,27]. The sample consisted of 799 patients, where 43% were female. The study by Zaghi et al. [23] had the highest number of participants (n = 348), and the studies by Govardhan et al. [20], Ferrés-Amat et al. [8] and Khan et al. [21] achieved the lowest sample size (n = 1). The mean age of the participants ranged from 17 days [8] to 79 years [23]. Zagui et al. [23] is the only study that includes patients over 65 years of age. The most frequently used classification for the diagnosis of frenulum has been the Kotlow classification [6,21,24,27], followed by the Coryllos classification [8,24,25] and the functional classification proposed by Yoon et al. [28] based on the tongue range of motion ratio [20,23].

In relation to the intervention protocols, only one article did not use lingual frenectomy [12]. The frenectomy technique varied between the articles using grooved probe and Metzenbaum dissecting scissors [22,23,25], with scissors and suture [23], with CO_2_ laser [6], Z-plastia [8] or with diode laser [27]. In all of the studies, MFT was used, although some of them put more emphasis on functional stimulation [6,8,20,22,23,24,25,27] and others on tongue-training exercises [12,21,26]. Speech therapy was also used in 36.4% of the articles [12,20,21,27] and breastfeeding sessions in 18.2% [24,25]. 

The exercises were performed once or twice daily [21] or three times a day [8,12,22]. Exercise repetitions were variable among the studies that specified them: 3–5 min bursts [21], eight exercises, each one 15 times [12] or two sequences of 15 repetitions [8].

MFT continued for 1 year or longer to prevent relapse of dysfunctional oral motor habits, to promote exclusive nasal breathing and to ensure long-term habituation of ideal resting oral posture.

Almost half of the studies did not specify the protocol used for MFT. In the studies in which it was explained, the treatment lasted 20 min twice a week [22] or 30 min [8,12,24,25]. In one of the articles [23], the myofunctional protocol incorporated bodywork, cranial therapy and/or myofascial therapy depending on the circumstance.

Treatment lasted about 4 weeks in most of the articles [8,21,22,23,24,25]. The longest total duration of intervention was achieved by Zagui et al. [23] (1 month of preoperative and 2 months postoperative MFT). There was another study in which TFM started before surgical treatment, but in this case it was one week earlier [8].

The variables studied were related to tongue mobility including the ankyloglossia grade evaluated by Kotlow, or the quick tongue-tie assessment tool [27], or tongue range of motion [12,20,26,27]. The strength and endurance of the tongue was also measured with the Iowa oral performance instrument (IOPIpro). In terms of functionality, it was evaluated with the assessment tool for lingual frenulum function, the Bristol tongue assessment tool and the degrees of lingual function [27] or by maximum interincisal opening with the tip of the tongue in contact with the maxillary incisor papillae (MOTTIP) [12,27] or maximum interincisal opening of the mouth (MIO) [12,26] or maximum opening mouth (MAB) [27], the lingual protrusion measured with orofacial myology [27].

Parameters related to respiratory issues such as mouth breathing during sleep [12,20,23], noisy breathing [20], sleep quality [6], snoring [23], obstructive sleep apnea syndrome (OSAS) measured by polysomnography (PSG) [27], airway and expiratory patency assessed by the protocol for the phonoaudiological assessment of breathing with scores (ProPABS) [12] and nasal or oronasal breathing analysis were evaluated [12].

In addition, there were also variables related to breastfeeding, such as weight gain [22,24,25], breastfeeding duration [22,24,25] and maternal pain measured by visual analogue scale (VAS) [22,24,25]. Clenching or grinding of teeth [23], ability to perform myofunctional therapy exercises [23], ease of swallow [23], speech sound production [6,20,21], satisfaction rate [23], quality of life measured with quality of life scale (QOL), myofascial tension [23], complications [8,23], perceived fatigue [8], pain evaluated by visual analogue scale (VAS) [8,25] or numerical rating scale (NRS) [27] were also assessed.

In terms of outcomes, there were improvements in speech sound production [6,20,21], breastfeeding duration [22,24,25], mother pain [22,24,25], weight gain [22,25], tongue mobility [12,20,22,26], strength and endurance of the tongue [12], obstructive sleep apnea syndrome [23,27], mouth breathing and snoring [20,23], quality of life [23], clenching teeth [23], myofascial tension [23] and pain after surgery [23].

The main characteristics of the studies are shown in Table 2.

### 3.3. Methodological Quality Assessment

The results of the quality assessment of the different studies are shown in Table 3, Table 4 and Table 5. Table 3 presents the methodological quality of the clinical trials. Table 4 and Table 5 show the methodological quality of the observational studies and the case studies, respectively.

The mean methodological quality of the clinical trials as measured by the PEDro scale was 5.6, that of the case studies as measured by the SCED scale was 3.3, and in the case of the observational studies 71% of the recommendations of the STROBE statement were met.

### 3.4. Risk of Bias of Included Randomised Clinical Trials

Regarding the risk of bias of the randomized clinical trials included in this review, the study conducted by Fioravanti et al. [27] presented the lowest risk of bias, as shown in Figure 2. It should be noted that the risk of bias is low in relation with attrition and reporting bias in all of them (Figure 3).

## 4. Discussion

This systematic review was developed with the aim of finding out the efficacy of MFT in people with ankyloglossia. In view of our results, we can conclude that there is not enough evidence that the MFT is effective on its own because in the articles evaluated it was accompanied by surgery. In addition, the studies were not of good methodological quality to draw extrapolatable conclusions. Even so, when both treatments have been combined, the results were positive.

Concerning characteristics of the participants, ankyloglossia has a high prevalence in infants aged 0–6 months, the population studied in 36% of the studies in our review, although the average age range is 8 years in 80% of the articles chosen. It is important to study this pathology at an early age in order not to condition the development of the cranio-mandibular-occlusal complex of the child and to avoid future problems when they reach adulthood [30].

According to the gender, in our review, the number of males is higher. Our results are in agreement with the data obtained in other research [31,32,33]. This may be due to the link of ankyloglossia to the X chromosome with variations in gene expression, in particular in the mutation of the T-box transcription factor gene (TBX22) during palatogenesis [34].

Another important aspect to consider is the heterogeneity in the scales for making the diagnosis of ankyloglossia. It should be unified on a more functional scale, such as the one proposed and validated by Yoon et al., after their analysis of 1052 subjects. It would be a more practical way to establish therapeutic objectives. This scale establishes 4 degrees of ankyloglossia: grade 1: tongue range of motion ratio is >80%, grade 2 50–80%, grade 3 <50% and grade 4 <25% [28].

In reference to the variables studied, a distinction could be made between those relating to babies and those studies in children or adults. Concerning infant studies, the most assessed has been nipple pain, feeding time and weight gain, as in the review by Walsh et al. [35]. Others are mostly found in adults, such as speech difficulties and upper respiratory tract development, as in the study by Jaikumar et al., (2022) in which aspects such as social interactions and academic activities are examined in depth. On the other hand, the variables studied in children and adults are speech difficulties and respiratory disturbances resulting from restricted mobility of the tongue. Nevertheless, social interactions and academic activities have not been studied as they were evaluated in the article of Jaikumar et al. [30].

Concerning the interventions, the option of frenectomy is the most commonly used method for lingual frenulum release in the neonatal and infant population and it is efficient for the improvement of symptoms caused by ankyloglossia [36]. Researchers of a systematic review of Shea et al., reported no serious complications; however, the authors added that the total number of infants studied was small as well as the number of trials which were also of low methodological quality [37]. The most frequently encountered complications have been pain, infections or hemorrhages [38].

Conservative treatment should therefore be taken into account [39], and the work of a multidisciplinary team would be essential [40]. Miranda et al., (2016) commented that MFT could be the only method of treatment in some cases and, in cases where surgery is necessary, it would be an essential part that could complement it. MFT can be given by both speech therapists and physiotherapists. The more functional part, which includes stretching, exercises and intrabuccal and extrabuccal massotherapy, would be given by a physiotherapist, and the area that involves more speech would be developed by a speech therapist [41].

Finally, in clinical settings, although MFT has been done with the surgery in ankyloglossia, there are therapeutic plots where this has been done without the support of surgery and with satisfactory effects: in preterm infants where myofunctional and orofacial therapy resulted in improvements in weight gain [42,43] or in obstructive sleep apnea [44] where decreases in apnea-hypopnea index by approximately 50% in adults and 62% in children with lowest oxygen saturations, snoring, and sleepiness outcomes improve in adults were found. Conversely, there has not been scientific evidence supporting the use of MFT in combination with orthodontic treatment to achieve better results in the correction of dentofacial disorders in orthodontic patients [45].

### Limitations and Future Recommendations

Some limitations should be remarked. The results provided by the present review should be taken with caution because of the limited number of controlled trials analyzed. Another limitation was the variability of the samples in the studies and the differences between the registered outcomes, which made it impossible to carry out a meta-analysis to complete the review.

There was a wide disparity in age range in the studies. Some were conducted in infants [22,24,25], in infants or children [6], others in children [8,12,20,26] or young adults [21] or even in people over 65 years of age [23].

Furthermore, the wide range measuring instruments used makes it difficult to compare the results on the studies and, in some of the trials [22,24,25], not only the patients were analyzed but also the maternal pain was the variable studied. With regard to MFT, many of the articles [6,20,27] did not describe the protocol accurately in terms of number of sessions, techniques used and time and frequency of treatment.

Despite an extensive search in six databases, without using date or language restrictions, only 11 articles were found, of which only three were randomized clinical trials and of these only one was of good methodological quality. However, as stated by Baxter et al. [6] (2020), it is difficult to conduct a randomized control group study as mothers of babies with ankyloglossia request surgery and benefit from it. In addition, 100% of the studies had an uncertain risk under other bias. There may be certain factors that could influence the results such as the assessment of certain variables that may be subjective or the different levels of compliance of patients or their relatives when performing the exercises at home.

Finally, a great limitation is that it was not possible to assess MFT without surgery being present because in all the articles frenectomy was performed. In only one article [12] the treatment did not incorporate surgery in the intervention protocol, but it was not specified whether it had been performed before.

Regarding future recommendations, it should be noted that myofunctional therapy could go beyond being an adjunct to surgery.

Frenectomy is an effective surgical treatment in people with ankyloglossia [27], but there will be cases where it is not necessary; however, there are not enough RCTs to draw firm conclusions [46]. It is therefore important that more studies on this pathology are done. A good diagnosis should also be made to differentiate those individuals with ankyloglossia who should go directly to surgery from those who could benefit from conservative treatment such as MFT [47]. Hence, it could be a great tool for individuals with ankyloglossia. Improvement should be measured at different levels according to the variables and age studied. Regarding the age range, it would be useful to have more information on how ankyloglossia develops in adults, as this pathology is associated with babies because of the problems it triggers in the breastfeeding period, but complications in adults have hardly been investigated [48].

In addition, pediatric and women’s physiotherapy has been booming in recent years, with the need for this type of therapy emerging as a solution to future problems in both the baby and the mother, not only functional but also social [4].

This systematic review could serve as a reference for future studies with a higher methodological quality that take into account the above-mentioned aspects.

## 5. Conclusions

According to the literature consulted in this review, surgery is more effective than myofunctional therapy, although better results are achieved if both are combined. Improvements have been found in babies and their mothers with regard to maternal pain, weight gain of babies and duration of breastfeeding. On the other hand, in children and adults it improves tongue mobility, strength and endurance, sleep apnea, mouth breathing and snoring, quality of life, clenching teeth, myofascial tension, pain after surgery and speech sound production.

These findings must be taken with caution because of the small number of articles and their quality. Future clinical trials using larger sample sizes and with higher methodological quality are needed. Overall, myofunctional therapy is expected to have a positive impact on patients with ankyloglossia.

## Figures and Tables

**Figure 1 ijerph-19-12347-f001:**
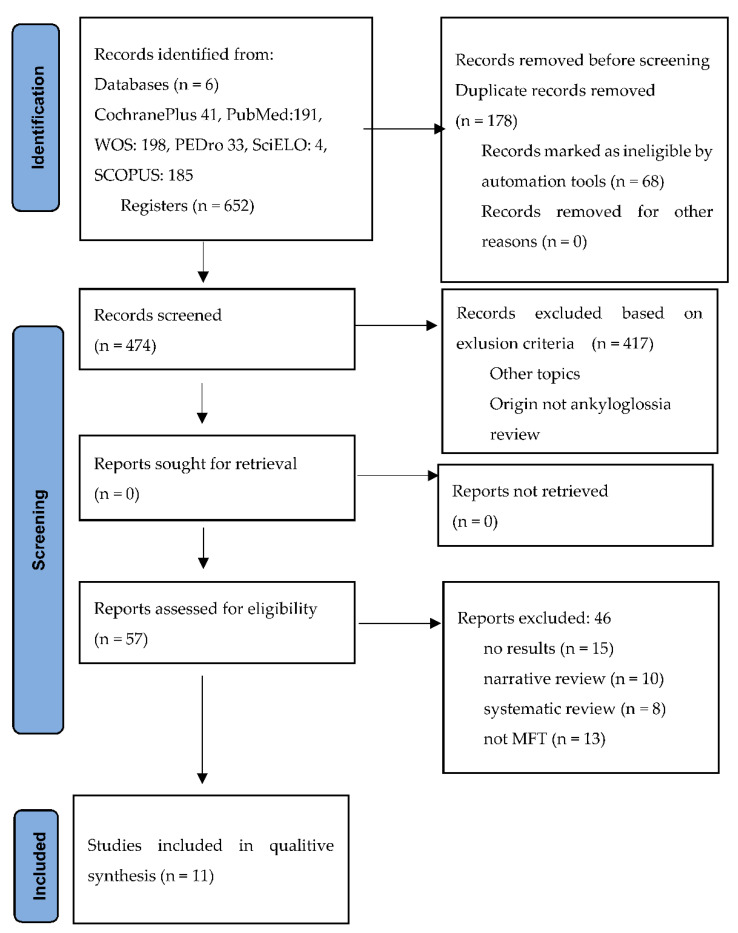
PRISMA 2020 flow diagram. From: Page MJ, McKenzie JE, Bossuyt PM, Boutron I, Hoffmann TC, Mulrow CD, et al. The PRISMA 2020 statement: an updated guideline for reporting systematic reviews. BMJ 2021;372: n71. doi: 10.1136/bmj.n71. For more information, visit: http://www.prisma-statement.org/ (accessed on 2 June 2022).

**Figure 2 ijerph-19-12347-f002:**
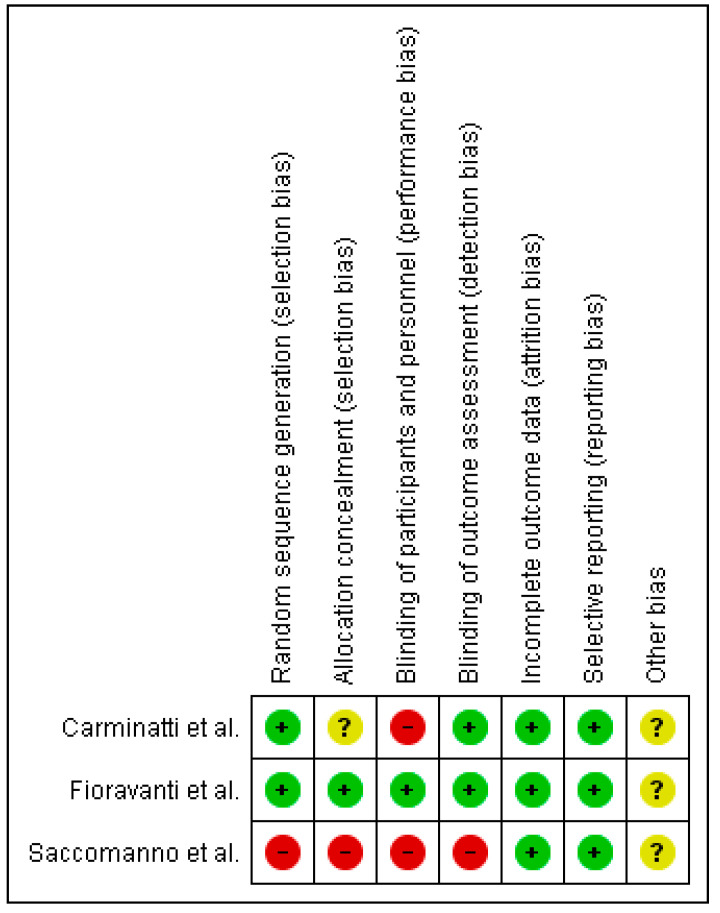
Risk of bias summary.

**Figure 3 ijerph-19-12347-f003:**
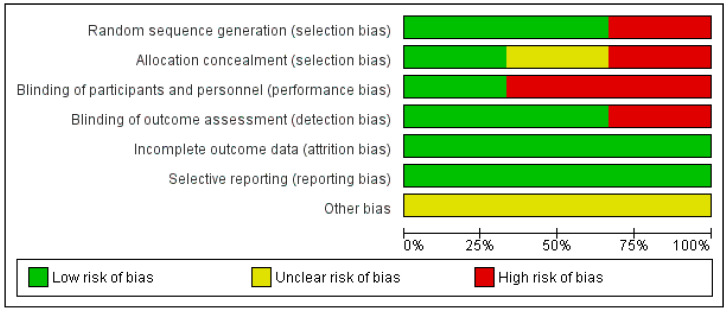
Risk of bias graph.

**Table 1 ijerph-19-12347-t001:** Search Combinations.

Databases	Search Strategy
Cochrane Plus	Ankyloglossia OR (tongue tie) AND physiotherapy in title abstract keyword
PubMed	(ankiloglossia OR tongue tie OR lingual frenum OR lingual frenulum OR (short lingual frenulum)) AND ((myofunctional therapy) OR (tongue orofacial exercises) OR (functional therapy protocol) OR (functional therapy) OR myofunctional OR (myofunctional training) OR (orofacial myology) OR (orofacial myofunctional therapy))
WOS	TITLE-ABS-KEY Ankyloglossia (topic) AND therapy (topic)
PEDro	Tongue therapy
SciELO	Tongue tie in all indexes
SCOPUS	TITLE-ABS-KEY (ankyloglossia OR (tongue AND tie) OR (lingual AND frenulum) OR (lingual AND frenum) AND therapy OR physiotherapy

**Table 2 ijerph-19-12347-t002:** Characteristics of the Study Intervention.

Author/ Year/Type of Study	Sample	Intervention	Outcomes/Measuring Instruments	Results
Govardhan et al., (2019) [20] case report	n = 1 3 years	LF + maxillary labial frenuloplasty + MFT + speech therapy	- Tongue range of motion - Speech sound production - Mouth breathing during sleep - Noisy breathing	- Improved speech sound production - Stopped mouth breathing and snoring while asleep - Eliminated tongue thrust
Ferrés-Amat et al., (2016) [22] case report	n = 1 17 days	LF with a fluted probe and Metzenbaum dissecting scissors + MFT	- Weight gain - Breastfeeding duration - Maternal pain (VAS)	- Improved breastfeeding duration, pain, weight gain
Khan et al., (2017) [21] case report	n = 1 20 years	LF + tongue training exercises + correction speech with speech therapist	- Speech sound production	- Improved speech sound production
Baxter et al., (2020) [6] observational	n = 37 13 months-12 years	LF with CO2 laser + MFT	- Speech, feeding and sleep (Likert scales)	- Improvements in: speech (89%), solid feeding (83%) and sleep (83%) reported by parents. - 50% speech-delayed children said new words (*p* = 0.008) - 76% slow eaters ate more rapidly (*p* < 0.001). - 72% restless sleepers slept less restlessly (*p* < 0.001).
Zagui et al., (2019) [23] observational	n = 348 28 months-79 years	LF with scissors, suture technique + MFT	- Satisfaction rate (survey) - Quality of life (QOL) - Mouth breathing - Snoring - Clenching or grinding of teeth - Myofascial tension - Complications/risks: pain	- Satisfaction rate: 91.1% - Improvements in quality of life: 87.4%, tongue mobility (96.5 ± 1.0%; clenching or grinding of teeth (91.0 ± 4.3%); ability to perform MFT exercises (89.8 ± 1.6%), easy of swallow (80.3 ± 3.5%, sleep quality (79.6 ± 2.6%), nasal breathing (78.4 ± 2.8%); neck, shoulder, facial tension or pain (77.5 ± 2.8); snoring (72.9 ± 3.4%); mouth breathing: 78.4%. - Complications: pain after surgery (45.1%), pain longer than 7 days (1.2%)
Ferrés-Amat et al., (2016) [8] observational	n = 101 4–14 years	LF + Z-plasty + MFT	- Ankyloglossia grade - Complications - Perceived fatigue	- Improvements in 96% patients (*p* = 0.001) - Correction (degrees 1 or 2): 29% (95% CI:20%, 38%) in first rehabilitation session - Correction (degrees 1 or 2): 96% (95% CI:90%, 98%) in last rehabilitation session - Complications in 6% patients (4 tongue bites, 1 hemorrhage, 2 infections)
Pastor-Vera et al., (2017) [24] observational	n = 61 0–6 months Group 1: 6 Group 2: 19 Group 3: 36	- Group 1: BFS - Group 2: MFT + BFS - Group 3: LF + MFT + BFS	- Weight gain - Pain - Breastfeeding duration	- Group 1,2 and 3: improvements in effectiveness and comfort of breastfeeding, with statisticalsignificance in group 2 (except type of BFS), in group 2 (except type of BFS) - Group 1, 2, 3: statistical significance in pain perceived by the mother
Ferrés-Amat et al., (2016) [25] observational	n = 171 0–6 months CG:n:33 IG_1_:n:50 IG_2_:n:88	G_1_: BFS G_2_: BFS + MFT G_3_: LF + BFS + MFT	- Pain (VAS) - Weight gain (before breastfeeding and stimulation session) - Breastfeeding duration	- Improvements in weight gain and breastfeeding duration - Improvements in pain: G_1_ 4.12 (2.67) vs. 0.70 (1.16)/G_2_ 5.10 (3.27) vs. 0.98 (1.46)/G_3_ 5.33 (3.07) vs. 0.81 (1.25)
Fioravanti et al., (2021) [27] RCT	n = 32 CG:16 8 severe OSAS IG:16 4–13 years 3 severe OSAS 13 moderate OSAS	CG: MFT + speech therapyIG: LF with diode laser	- Pain (NRS) - OSAS (PSG) - Anquiloglossia grade (Kotlow, quick tongue-tie assessment tool) - MAB (oralfacial myology) - MOTTIP (oralfacial myology) - Lingual protrusion (oralfacial myology) - Functional assessment (assessment tool for lingual frenulum function, Bristol tongue assessment tool, degrees of lingual function)	- No significant differences between the groups in Kotlow (U = 99.8; *p* = 0.270), MAB (U = 106.5; *p* = 0.407), MOTTIP (U = 116; *p* = 0.649) and protrusion (U = 119.5; *p* = 0.747) - Improvements IG: Kotlow (Z = −3.521; *p* < 0.001), MAB (Z = −3.436; *p* < 0.01), MOTTIP (Z = −3.536; *p* < 0.001) and protrusion(Z = −3.527; *p* < 0.001). - Improvements CG: Kotlow (Z = −3.531; *p* < 0.001), MAB (Z = −3.088; *p* < 0.01), MOTTIP(Z = −3.412; *p* < 0.01) and protrusion (Z = −3.426; *p* < 0.01). - OSAS: IG 93.8% mild OSAS; 6.2% moderate OSAS vs. CG 18.75% mild OSAS; 62.5% Moderate OSAS;18.74% Severe OSAS
Saccomanno et al., (2019) [12] RCT	n = 6 4.5–11.7 years CG:2 IG_1_:2 IG_2_:2	- CG: no therapy - IG_1_: MFT + home exercises without speech therapist’s supervision. - IG_2_: MFT + home exercises with speech therapist´s supervision	- IOPI pro: tongue and lip strength and endurance - ProPABS: airway and expiratory airway patency - Clinical examination of saliva, water and biscuit swallowing - Range of motion: TRMR, MOTTIP, MIO. - Nasal or oronasal breathing analysis	- CG: no results detected - IG_1_: no results highlighted - IG_2_: positive results in tongue and lip strength and endurance and range of motion.
Carminatti et al., (2013) [26] RCT	n = 40 CG:20 IG:20 6–12 years	- IG: LF + isotonic tongue exercises - CG: LF + no treatment	- Tongue mobility - MIO - MIO with the tip of the tongue touching the incisive papilla	- EG improved tongue mobility (*p* = 0.016) and - MIO (*p* = 0.024)

LF: lingual frenectomy; MFT: myofunctional therapy; QOL: quality of life scale; VAS: visual analogue scale; BFS: breastfeeding; RCT: randomized controlled clinical trial; CG: control group; IG: intervention group; BFS: breastfeeding; NRS: numerical rating scale; PSG: polysomnography; OSAS: obstructive sleep apnea syndrome; Kotlow: classification frenulum; MAB: maximum opening mouth; MOTTIP: maximum interincisal opening with the tip of the tongue in contact with the maxillary incisor papillae; IOPI: Iowa oral performance instrument; ProPABS: protocol for the phonoaudiological assessment of breathing with scores; TRMR: tongue range of motion; MIO: maximum interincisal opening of the mouth.

**Table 3 ijerph-19-12347-t003:** Quality of Clinical Trials measured with the PEDro Scale.

Author, (Year)	Item 1	Item 2	Item 3	Item 4	Item 5	Item 6	Item 7	Item 8	Item 9	Item 10	Item 11	Total
Fioravanti et al., (2021) [27]	1	1	1	1	0	1	1	1	1	1	0	9/10
Saccomanno et al., (2019) [12]	1	0	0	1	0	0	0	1	1	0	0	3/10
Carminatti et al., (2013) [26]	1	0	0	1	0	0	1	1	1	1	0	5/10

**Table 4 ijerph-19-12347-t004:** Quality Assessment of Observational Studies using the STROBE Statement [29].

Evaluated Section	Item	Baxter et al., (2020) [6]	Zagui et al., (2019) [23]	Ferrés-Amat et al., (2016) [8]	Pastor-Vera et al., (2017) [24]	Ferrés-Amat et al., (2016) [25]
Title and abstract	1	✓	✓	✓	✓	✓
I: context	2	✓	✓	✓	✓	✓
I: objectives	3		✓	✓	✓	✓
M: study design	4	✓	✓	✓	✓	✓
M: context	5	✓	✓	✓	✓	✓
M: participants	6	✓	✓	✓	✓	✓
M: outcomes	7	✓	✓	✓	✓	✓
M: data sources/measures	8		✓	✓	✓	
M: biases	9					
M: sample size	10					
M: quantitative variables	11	✓		✓		✓
M: statical methods	12	✓	✓	✓	✓	✓
R: Participants	13		✓	✓	✓	
R: descriptive data	14		✓	✓	✓	✓
R: outcome of variable data	15	✓	✓	✓	✓	✓
R: main results	16	✓	✓		✓	✓
R: other analyses	17	✓				
D: key results	18	✓	✓	✓	✓	✓
D: limitations	19	✓	✓		✓	
D: interpretation	20	✓	✓	✓	✓	✓
D: generalizability	21	✓				
D: Other information: financing	22	✓	✓			

Introduction; M: material and methods; R: results; D: discussion.

**Table 5 ijerph-19-12347-t005:** Quality of the Case Studies, as measured by the SCED Scale.

Author, (Year)	Item 1	Item 2	Item 3	Item 4	Item 5	Item 6	Item 7	Item 8	Item 9	Item 10	Item 11	Total
Govardhan et al., (2019) [20]	1	1	1	1	0	0	0	0	0	0	0	3/10
Ferrés-Amat et al., (2016) [8]	1	1	1	1	0	1	0	0	0	0	0	4/10
Khan et al., (2017) [21]	1	1	1	1	0	1	0	0	0	0	0	3/10

## Data Availability

Not applicable.

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
