# Peer review of "Effectiveness of Myofunctional Therapy in Ankyloglossia: A Systematic Review"

_ijerph, 2022, doi:10.3390/ijerph191912347_

Round 1
Reviewer 1 Report
The purpose of the study was to the scientific evidence and assess its quality regarding the use of myofunctional therapy in synthesize ankyloglossia being selected 11 studies for this purpose. As conclude by the authors, myofunctional therapy achieves improvements in maternal pain, weight gain of babies, duration of breastfeeding, tongue mobility, strength and endurance, sleep apnea, mouth breathing and snoring, quality of life, clenching teeth, myofascial tension, pain after surgery and speech sound production. It cannot be considered more effective than surgical intervention.
The study is interesting, the title reflects the content, the aim is clear and the methods seem appropriated. The study limitations are also explained extensively and clearly. However, I have some concerns to be clarified or modified.
- I suggest to perform a revision in some English “terms”: “synthesise” replace by “synthesize”; “sleep apnoea” replace by “sleep apnea”……
- In the abstract, the authors conclude that the better results have been achieved if surgery was complemented by myofunctional therapy but after described that due to the small number of trials and their quality, the results cannot be conclusive. Future clinical trials using larger 27 sample sizes and with higher methodological quality and I suggest to clarify the sentences that can lead to a misunderstanding;
- In the PRISMA 2020 flow diagram, please include the reason for exclusion after the first screening (417);
- Line 157: 36.4 refers percentage?
- In the Table 2. Characteristics of the Study Intervention only the gender of the sample (n=1) for the first case report is described.
- There is an error in the table number since there are two tables 2, please correct. In the second table 2 (Quality of Clinical Trials measured with the PEDro Scale) “ítem” must be replaced by “item”.
- The same word (“ítem”) must be corrected in the table 4);
- A explanation of figure 3 must be included.
- Please, revise the final conclusion as suggest for the abstract section.
Author Response
Dear Editor and reviewers of the manuscript entitled “Effectiveness of Myofunctional therapy in Ankyloglossia: a systematic review”.
First of all, we would like to thank you for your comments and for allowing us to address the issues you raise to improve the manuscript’s quality. We appreciate your observations and the time devoted to the constructive criticism and feedback of our manuscript. Please find the answer to your comments below and the recommended changes have been highlighted in yelow in the manuscript.
Response to reviewer 1 Coments.
Point 1. I suggest to perform a revision in some English “terms”: “synthesise” replace by “synthesize”; “sleep apnoea” replace by “sleep apnea”……
Response 1. We have revised the entire manuscript to correct these terms. Thank you very much for make better our manuscript.
Point 2. In the abstract, the authors conclude that the better results have been achieved if surgery was complemented by myofunctional therapy but after described that due to the small number of trials and their quality, the results cannot be conclusive. Future clinical trials using larger 27 sample sizes and with higher methodological quality and I suggest to clarify the sentences that can lead to a misunderstanding.
Response 2. In order to make our manuscript easier to understand this issue has been modified in conclusion section. Thank you for your careful review.
Point 3. In the PRISMA 2020 flow diagram, please include the reason for exclusion after the first screening (417);
Response 3. The exclusion reason has been added.
Point 4. Line 157: 36.4 refers percentage?
Response 4. We are very thankful with the improvements proposed by the reviewer. Indeed, we were referring to the percentage. Percentage has been added.
Point 5. In the Table 2. Characteristics of the Study Intervention only the gender of the sample (n=1) for the first case report is described.
Response 5. We have removed this data from the table to make it more uniform. Thank you for your consideration.
Point 6. There is an error in the table number since there are two tables 2, please correct. In the second table 2 (Quality of Clinical Trials measured with the PEDro Scale) “ítem” must be replaced by “item”.
Response 6. These changes have been made. Thank you for improving our manuscript.
Point 7. The same word (“ítem”) must be corrected in the table 4);
Response 7. These changes have been made. Thank you for your comment.
Point 8. An explanation of figure 3 must be included.
Response 8. The explanation has been included in page 9, lines 226
Point 9. Please, revise the final conclusion as suggest for the abstract section.
Response 9. In order to make our manuscript easier to understand this issue has been modified in conclusion section. Thank you for your careful review.
Reviewer 2 Report
This article is a systematic review deals with effectiveness of myofunctional therapy (MFT) in ankyloglossia. The study highlights that the best clinical results have been achieved if surgery was combined with MFT. The article is well written and interesting. However, there are some minor issues that need to be considered.
- The authors did not describe and distinguish different surgical procedures used in ankyloglossia surgery, such as: frenotomy, frenulectomy, frenuloplasty, miofrenuloplasty, Z-plasty, and V-Y plasty (for example in: Miofrenuloplasty for full functional tongue release in ankyloglossia in adults and adolescents—Preliminary report and step-by-step technique showcase)
- In methodology the authors did not mention the range of time period of evaluating articles
- In line 155 and Table 2.- should be Z-plasty instead of “plastia in Z”
- In line 230- should be “MFT”
- In line 157 probably “%” is missing
Author Response
Dear Editor and reviewers of the manuscript entitled “Effectiveness of Myofunctional therapy in Ankyloglossia: a systematic review”.
First of all, we would like to thank you for your comments and for allowing us to address the issues you raise to improve the manuscript’s quality. We appreciate your observations and the time devoted to the constructive criticism and feedback of our manuscript. Please find the answer to your comments below and the recommended changes have been highlighted in yelow in the manuscript.
Response To Reviewer 2 Coments
Point 1. The authors did not describe and distinguish different surgical procedures used in ankyloglossia surgery, such as: frenotomy, frenulectomy, frenuloplasty, miofrenuloplasty, Z-plasty, and V-Y plasty (for example in: Miofrenuloplasty for full functional tongue release in ankyloglossia in adults and adolescents—Preliminary report and step-by-step technique showcase)
Response 1. Thanks for your careful review. We have added the different surgical procedures in page 2, line 50. Thanks you for recommending the article.
Point 2. In methodology the authors did not mention the range of time period of evaluating articles.
Response 2. Thank you for your comment. In the methodology section, the range of time period has been added. Page 13.
Point 3. In line 155 and Table 2.- should be Z-plasty instead of “plastia in Z”
Response 3. We are so grateful with your appreciation.
Point 4. In line 230- should be “MFT”.
Response 4. The error has been corrected. Thank you for your careful review.
Point 5. In line 157 probably “%” is missing.
Response 5. We are very thankful with the improvements proposed by the reviewer. Indeed, we were referring to the percentage. Percentage has been added.